# A New Strategy for Animal Research: Attending to Dissent

**DOI:** 10.3390/ani13091491

**Published:** 2023-04-27

**Authors:** Jane Johnson

**Affiliations:** 1Department of Philosophy, Macquarie University, Sydney, NSW 2109, Australia; jane.johnson@mq.edu.au; 2Macquarie University Research Centre for Agency, Values and Ethics, Macquarie University, Sydney, NSW 2109, Australia

**Keywords:** animal research, animal ethics, animal research ethics, dissent

## Abstract

**Simple Summary:**

Concepts typically used for human research ethics are beginning to be applied to animals in research. This paper focuses on the concept of participant dissent, and examines the options researchers can pursue and the consequences of these options, if researchers take the dissent of research animals seriously.

**Abstract:**

Increasingly, ethical concepts ordinarily reserved for the human research setting have been applied to nonhuman animals in research. This comes at the same time as concerns mount over challenges in translating the results of biomedical research with animals to human clinical benefit. This paper argues that applying the concept of dissent derived from research with humans to the context of animals can help to address a number of these translational issues, thereby providing an epistemological reason to take animal dissent seriously. This epistemological rationale can be added to the practical and ethical reasons for attending to animal dissent. Having made a case for recognizing the dissent of animals in biomedical research, the consequences that follow from this for the conduct of research are discussed. If animal researchers attend to dissent, then it seems that there are three types of strategy available: to override dissent, to train animals in such a way as to circumvent potential dissent, or to alter how research is conducted in order to be responsive to dissent. Only this last option has the potential to address all the types of reasons that motivate us to take dissent seriously; however, this would involve a significant reshaping of the practice of animal research.

## 1. Introduction

Around the globe, the early 2020s have seen significant changes regarding animal research. In 2021, the European Parliament voted in support of a resolution to phase out the use of animals in research, testing, and education by adopting an action plan [1], while in the United States 2022 saw the passing of the Food and Drug Administration (FDA) Modernization Act, which ended the FDA requirement that experimental drugs be tested on animals prior to their use in human clinical trials [2] These changes have been underpinned by a number of factors. For instance, there is growing acknowledgment that preclinical research on nonhuman animals intended to deliver human clinical benefit has not delivered on its promise [3,4,5], with estimates suggesting that only between 5% and 10% of animal studies translate to successful interventions for human patients [6]. The changing regulatory environment has also been enabled by the rise of new non-animal approach methodologies (NAMs) with the potential to better predict clinical responses in humans compared to existing animal models. Although there are barriers to developing and implementing non-animal alternatives (e.g., due to inertia, a lack of knowledge, existing research cultures, etc.) [1,4], if these barriers are recognized and addressed, non-animal alternatives can assuage concerns that human safety will be compromised and medical progress stymied without animal models.

This shift in how animals in research are regarded comes at the same time as parallels have increasingly been drawn between approaches to human research ethics and what has come to be labelled ‘animal research ethics’ [7]. Concepts previously considered the sole preserve of research with humans have begun to be used in the context of nonhuman animals. Of particular interest in this paper is the concept of dissent. This paper will argue that attending to dissent has the potential to help address a number of the epistemological and ethical challenges presented by biomedical research with animals, as well as certain of its practical ones. Having defined dissent and made a case for attending to animal dissent in biomedical research, the consequences for the conduct of animal research are systematically examined. If animal researchers attend to dissent, it seems that there are three strategies available to them: to override dissent, to train animals in such a way as to circumvent their potential dissent, or to alter how research is conducted in order to be responsive to dissent. Only this last option has the potential to comprehensively address all the types of reasons that motivate taking dissent seriously; however, this would involve a significant reshaping of the practice of animal research. 

## 2. Consent, Assent, Acquiescence, and Dissent

To appropriately situate dissent, it is helpful to define several related concepts. In the context of human research, informed consent has received significant and sustained consideration over many decades. However, for the purposes of the discussion here, a brief indication of what it involves will suffice. In the context of research, informed consent involves a voluntary agreement to participate on the part of the participant, secured either verbally or in writing, following an explanation of what the research entails. Informed consent can be revoked at any time during research [8]. If consent cannot be secured because a human lacks capacity, another party may be nominated to make decisions on behalf of that individual. This guardian or surrogate can provide or withhold consent regarding participation, possibly in tandem with the assent of the research participant. 

Assent indicates a preparedness to be involved in research, and may be solicited in a number of ways. The protocol and what it involves ought to be communicated to the potential participant in an appropriate manner, which could include via demonstration. Thus, understanding of the research is required, though there is no general agreement over the extent of understanding required in order for assent to be deemed legitimate [9]. An indication of agreement is then sought from the research subject, which might be expressed verbally, and/or there may be some interpretation of behaviour that supports assent. 

Unlike consent and assent, the related concept of acquiescence does not require a participant’s prospective understanding of what research entails. Acquiescence to research simply involves cooperation and an absence of dissent [10].

Dissent is less demanding than consent or assent, and is therefore applicable in the context of children, who are incapable of consent or assent, and (as will be argued here) animals. As Fenton states, dissent "does not require sophisticated or complex awareness; it is enough that a procedure is distressing or that the relevant individual does not want to co-operate" [11] (p. 401); I disagree with Fenton, however, that dissent requires the capacity ‘to anticipate the future occurrence of distress, pain, or stress’ (p. 401), as one can dissent to occurrent distress, pain, or stress. Similarly, Kantin and Wendler’s account of dissent can apply to animals since they regard dissent as "an expressed objection," with ‘objection’ interpreted broadly [10]. However, dissent is not limited, in the way these authors suggest, to ‘what an individual is experiencing as a result of a research procedure,’ as one could prospectively dissent to research in which one chose not to be enrolled. 

Dissent can be expressed verbally or through behaviour, which might include avoiding contact with researchers, vocalizing loudly, or trying to escape. Precisely what this behaviour means, however, involves interpretation that is fallible. Reasons unrelated to the research might be driving the behaviour, in which case the behaviour might not represent genuine dissent. For example, the individual might be tired or hungry, or dislike the researcher, rather than be expressing dissent to the research itself. Unlike consent, dissent does not demand an understanding of what a research protocol involves, and dissent can be legitimately overridden in certain cases by a party who does understand the nature of the research. For instance, a child might dissent to research with the potential for therapeutic benefit which is unable to be accessed by other means. In such a case, a parent might ignore the child’s dissent on the grounds that what they regard as a greater harm (such as morbidity or mortality) might be avoided through research participation [12].

Dissent can be expressed at any point in research, though the point at which it arises may limit the nature of what can be done in response. For instance, there may be a ‘point of no return’ in a protocol (e.g., the drug has been administered, which then causes the dissent) such that nothing can be altered in response to the dissent, at least with respect to that individual enrolled in that research protocol. The dissent might, however, motivate changes in future research. 

Although the terms outlined above (especially consent and assent) are commonplace in discussions of human research, they are almost entirely absent in the context of research with nonhuman animals. This is likely for a variety of reasons, including that it is simply assumed that nonhuman animals are incapable of the kind of prospective understanding of what research entails that is required, by definition, for consent and assent. 

There are, however, a number of recent exceptions to this general neglect in the philosophical and bioethical literature in which animal research is considered. For instance, in addition to those authors noted above (Fenton; Kantin and Wendler), Ferdowsian and colleagues canvass the applicability of assent, dissent, and surrogate decision-makers in the context of research with animals [13]. Healy and Pepper argue that nonhuman animals can express their will through acts of both assent and dissent [14], while Arnason argues that although animals can neither consent or assent to research (though certain great apes may be an exception with respect to assent), they may be able to acquiesce or dissent [7]. In previous work, a collaborator and I articulated an approach to biomedical research modelled on veterinary clinical research, according to which each animal has a human who can consent on the animal’s behalf [15].

Outside the philosophical and ethical literature, there has been sporadic engagement with these concepts in the context of animal research. In research regulation, for instance, the 2nd Recommendation of the 2011 report by the Institute of Medicine in the United States into the use of chimpanzees in biomedical and behavioural research stated that comparative genomics and behavioural research in chimpanzees should be limited to studies which are ‘performed on acquiescent animals’ [16].

Beyond the biomedical context, other types of research with animals have made appeals to concepts related to willingness to participate in research. For instance, in research undertaken on canines at Emory, researchers sought what they referred to as ‘assent’ from the companion animals they recruited, in addition to owner consent. The research involved taking functional MRIs of the dogs; during operation of the scanner, dogs were unrestrained and able to exit at will. Assent was regarded as being demonstrated when an animal did not exhibit ‘any signs of not wanting to participate’ [17]. It should be noted this use of assent differs from the above definitions, and appears closer to acquiescence than assent, as assent demands at least some prospective understanding. 

Another instance of research that factors in the willingness of animals to participate comes from work with chimpanzees at Tchimponga Sanctuary in the Republic of Congo and bonobos at Lola ya Bonobo in the Democratic Republic of Congo intended to compare the developmental stages of physical cognition in children and animals of comparable age. The chimpanzees and bonobos participated in computerized tests in their dormitory, and were free to terminate the test at any time and to leave the testing area [18]. A further example involves research currently being undertaken at Farm Sanctuary in New York, which follows a set of guidelines that include a requirement for consent when research requires active participation by experimental subjects. Consent is established through ‘voluntary participation in the research and careful observation of body language’ ([19], p. 3). Again it should be noted, this definition of consent divergences from the definitions generally provided in the literature on human research ethics and articulated in this paper.

Although examples such as those identified above can be found, appeals to concepts such as consent, assent, acquiescence, and dissent are not the norm in biomedical research with animals. This paper seeks to make progress on this omission by focusing on dissent. This is partly because dissent has the potential to apply more broadly than consent or assent to animals in research, since, amongst other features, it does not demand high levels of awareness or cognition. The most commonly used research animals (rodents and zebrafish) may be able to express dissent in spite of being unable to provide consent or assent. As the evolutionary biologist Brian Hare has argued, “[m]ost animals are capable of indicating behaviorally that they do not want to participate in an activity” [20].

The discussion of dissent provided below begins with an outline of the different types of reasons that dissent in animal research merits attention, before canvassing how it might be addressed.

## 3. Reasons to Attend to Dissent

### 3.1. Practical

The practical implications of dissent are self-evident. Dissent may be exhibited physically, manifesting in resistance, attempting to escape, lashing out, etc. Such noncompliance might be met with force or with physical or chemical restraints. In all these cases there is a heightened risk of injury and compromised safety for the researcher, the research subject, and other proximate humans and animals. 

### 3.2. Epistemological

Dissent has the potential to impact research outcomes (which in turn can influence translation to humans) in two main ways: by preventing data collection, and by skewing results. In the first case, an animal’s dissent might present such that it effectively amounts to insurmountable resistance, making it impossible to perform the intended protocol and resulting in no data being collected. In the second, results can be distorted in a number of ways. Dissent is likely to be associated with stress, anxiety, and fear, which can impact physiology and the reliability of any data obtained [21]. For example, in rodents, stress can lead to elevated cortisol levels and dampened pain responses, which affect behaviour and confound results [22]. If dissent manifests such that physical or chemical restraint is required, this may have implications for future research. For instance, use of tranquilizers and other restraints is associated with the development of behavioural disorders in chimpanzees [9]. Unless a future protocol is investigating such a behavioural disorder, results obtained from these animals may be unreliable. 

### 3.3. Ethical 

Ethical reasons can support the argument for attending to dissent. These reasons can be formulated in a number of ways depending on the ethical approach being adopted. For instance, advocates of animal rights might express worries about disregarding animal agency, while consequentialists might mount concerns grounded in impacts on welfare. The approach I support appeals to a conceptualization of animals as vulnerable subjects. Although this position has been argued for in greater depth elsewhere [23], in order to ground these claims here several of the main points are briefly summarized below. 

Animals involved in research experience what Mackenzie, Rogers, and Dodds label in the context of human research ethics "inherent, situational, and pathogenic" vulnerabilities [24]. Pain, suffering, death, etc., are inherent vulnerabilities shared by all biological creatures, while situational vulnerabilities arise from the personal, social, political, economic, or environmental circumstances of individuals or groups. In research, animals are made situationally vulnerable through their institutional setting, the purpose of their existence, and their dependence on humans to meet their needs. The precarious nature of their situation means that otherwise latent inherent vulnerabilities may be realized. 

Pathogenic vulnerabilities form a subset of this situational category, and are brought about either through morally dysfunctional relationships or when attempts to remedy existing vulnerabilities actually worsen them or introduce new vulnerabilities. Nonhumans in research can be construed as pathogenically vulnerable in both these ways, including as a result of the activities of Animal Ethics Committees (AECs) [23].

A fundamental principle of research ethics is that the vulnerable ought to be protected [25]. Therefore, if animals are regarded as inherently, situationally, and pathogenically vulnerable in research, and if recognizing their dissent protects them in some way (e.g., by either removing or ameliorating their vulnerability), this provides another type of reason to take dissent into account. 

The argument in this paper is for a vulnerability approach to animals in research. However in order to understand how the strategies canvassed below might operate in the current ethical environment of animal research, there will be some discussion of the consequentialist ethics which dominates contemporary research practice. While animal suffering and welfare are both morally important in this consequentialist framework, they are subordinate to human interests. This approach is operationalised via the ‘three Rs’, which propose the Replacement of animal research with alternatives where possible, Reduction in the number of animals used in experimentation consistent with achieving desired results, and Refining techniques to minimise animal pain and suffering [13]. On this approach, dissent can be considered under Refinement. 

The ethical reasons described in this paper are those that bear directly on nonhuman animals. However, a case can be made that there are ethical reasons to attend to animal dissent based on the impact of such dissent on those humans who care and work with research animals. This aligns with increasing acknowledgement of the challenges and potential negative effects on individuals who work closely with such animals [26,27].

With these three kinds of reasons supporting consideration of dissent (i.e. practical, epistemological and ethical), the next section articulates different options available to animal research workers (i.e., animal researchers, laboratory technicians, and animal carers) in response to dissent on the part of the animals in their care. 

## 4. Strategies in the Face of Dissent

If dissent is to be explicitly addressed, it needs to be identified. This imposes requirements on animal research workers which can be demanding and outside their normal remit. For example, heightened attentiveness to animals may be required to proactively seek out signs of dissent. Further, there are both species-specific and individual manifestations of dissent. For instance, prey species may exhibit stress by freezing, whereas a predator may become more aggressive. As studies into animal personalities demonstrate, there are variations across species that affect how individual animals might experience research [28]. This impacts whether or not they might dissent to protocols and how this dissent might manifest, requiring a nuanced knowledge of individual animals. Finally, in order to respond appropriately, animal research workers need to able to discern whether behaviour represents genuine dissent or has a different source.

Again, it may be challenging and time-consuming to ascertain the causal chain that creates a response which appears to be dissent. All these factors mean that it may be difficult to recognize dissent, and in spite of their best efforts, those charged with identifying dissent may not always get it right. 

### 4.1. Overriding Dissent

One possible strategy in the face of dissent is to override it. This is different to simply ignoring dissent, even though the outcome may be effectively the same. In the case of overriding dissent, the dissent has been recognised and given consideration before a judgement is made to overrule it in this particular case. 

With respect to addressing the three types of reasons to attend to dissent, overriding dissent has limitations. The practical challenges to running experimental protocols persist, as do concerns that resulting data will be skewed. Yet, overriding dissent might nonetheless address ethical worries. Whether or not this is the case depends on the details of particular instances, i.e., the reasons for overriding dissent and the broader circumstances in which it occurs, as will be explained below. 

The effects of the inherent vulnerability experienced by research animals might actually be alleviated in certain cases when dissent is overridden. For instance, research might offer therapeutic benefits by treating a disease or condition from which an animal suffers. Even if the vulnerability is a product of the experimental protocol (i.e., the disease or condition was induced as part of the research), treating it could be beneficial. In this case, the consequentialist too has reasons to support overriding dissent. 

However, if overriding dissent contributes to or exacerbates situational vulnerability, then it is problematic. Considering the at least surface similarities between not attending to dissent at all and overriding it, it seems plausible that overriding dissent risks reinforcing the instrumental use of animals in research, i.e., seeing animals as tools whose compliance can be compelled in the pursuit of research goals. This in turn is linked to concerns about compromising their agency and the adverse impacts that flow from it. Agency is important to many animals; overriding their dissent may represent a harm [10,29], and could in fact lead to learned helplessness on the part of animals. Frequently, research animals have limited ways of meaningfully impacting their situation, meaning that recognizing rather than ignoring their dissent in the research context can be important.

Overriding dissent generates ethical concerns for the consequentialist too. Dissent (which may, at a minimum, represent an expression of discomfort or distress on the part of an animal and is likely to be associated with elevated levels of stress, anxiety, and fear) reflects an animal’s welfare being adversely affected, in which case overriding dissent might constitute a missed opportunity to better understand and address that welfare issue. 

Concerns that disregarding dissent might represent a failure to acknowledge welfare issues contributes to a worry that overriding dissent could lead to further and significant harms. That is, dissent might be an early indicator of the potential for an animal to develop longer-term trauma such as post-traumatic stress disorder (PTSD), depression, etc. [30].

### 4.2. Training

Training is another strategy that responds to dissent, though in this case it is prospective and intended to circumvent anticipated or possible dissent. For instance, chimpanzees and macaques can be trained to present their arms for the extraction of blood, a procedure which can otherwise be challenging to perform and which might generate dissent [31].

If successful, this strategy overcomes several of the practical challenges associated with dissent. It seems likely that training would assuage certain of the epistemological worries around dissent as well. Trained animals could enable protocols to be undertaken. Furthermore, stress, which can be a major factor in skewing research results, can be assuaged, as evidence from animals trained to accept husbandry procedures in zoos indicates that they experience reduced behavioural stress [32].

In terms of inherent vulnerability, training appears to represent an ideal way of addressing ethical concerns, particularly for those who work in animal research. If animals are not exhibiting dissent and associated stress, both their short-term and longer-term welfare is improved. This is because disorders such as PTSD develop based on earlier trauma, and if there is no earlier trauma, then PTSD is prevented. The success of this approach is of course predicated on the training itself not generating welfare problems. One concern that does remain is that this approach may mask issues with animal welfare that dissent might signal. For instance, while a trained animal might not exhibit dissent as a manifestation of problems experienced with their welfare, this does not mean that no breach of welfare is occurring. 

Although training may address issues around inherent vulnerability, it may introduce new situational and even pathogenic vulnerabilities. Training attempts to change the situation of animals, sometimes in ways that limit agency, i.e., by effectively altering the range of choices available in certain situations, potentially shaping behaviour in ways that run counter to the interests of animals in order to facilitate compliance with the interests of research and researchers. It is worth noting that it might be possible to use training to extend agency as well, at least in certain respects. For instance, animals might be trained in such a way that they are encouraged to explore new opportunities which expand the choices available to them.

Even though training may be a well-intentioned mechanism for achieving the good end of limiting fear, anxiety, and stress in particular research animals, it may nonetheless worsen the situation of research animals in general by facilitating the status quo. With appropriate training in place, the practice of research can continue, since suffering is minimized and dissent is not expressed. Thereby, through shifting attention away from what might otherwise be in the foreground, training may forestall broader alteration of research practices. To expand on this potentially counter-intuitive point, I draw a comparison to Temple Grandin’s work. 

Grandin has contributed to improvements in the design of abattoirs such that animals heading to slaughter experience less fear, stress, and anxiety, i.e., their suffering is minimized. On the face of it, this seems like a good thing. However, what this work simultaneously does is enable the practice of industrial animal agriculture to continue, by appearing to achieve more than it does. Although minimizing the suffering associated with animal death is important, it is just one of a nexus of issues, some of them more fundamental than others. For instance, there are substantive ethical issues involving the conditions in which production animals are bred and live, environmental concerns, worries about the instrumental use of animals, etc., which are not addressed by changes to the setting in which animals are killed. 

In contrast, for the consequentialist, training appears to be an ideal strategy for circumventing dissent and delivering good outcomes for research animals. It is surely preferable, in this view, that the negative experiences of animals involved in research be minimized, provided that this does not impede the overarching goals of the research protocol. In general, it would appear that pre-emptive and positive training which creates less stress, better welfare, and the potential for exploring new opportunities supports good outcomes for research animals.

### 4.3. Changing Research

The final strategy considered here involves altering research in response to dissent. This strategy promises to answer the practical and epistemological worries around dissent as well as certain ethical challenges.

To be responsive to dissent, its presence needs to be detected; as indicated above, determining whether it is genuinely present is challenging and the process can be fallible. Identifying dissent demands a set of skills on the part of those working with research animals which may be outside their ken, along with a degree of attentiveness to research animals which may be potentially harmful [26,33].

Setting aside the challenges associated with identifying dissent, if dissent is present, one response is to alter its cause to allow the research to proceed. For instance, rodents struggling when administered pharmaceuticals via oral gavage could instead receive an edible gel. The practical advantage of this change is clear, as are the attendant epistemological advantages. Oral gavage impacts cardiovascular parameters, with associated increases in blood pressure, heart rate, and complications; consumption via gel does not result in these changes [34]. There are welfare benefits in terms of stress and the likelihood of injury, as oral gavage can lead to rupture of the oesophagus or stomach, or inadvertent injection of chemicals into the lungs. Yet, despite these benefits, changing the research protocol is not straightforward. Altering approved methods often requires reassessment by an AEC, who may legitimately question why researchers previously proposed a method which turned out to be problematic. Presumably, however, researchers would incorporate the knowledge gained from this experience into the design of future protocols.

Another way in which research can demonstrate responsiveness to dissent is by only being undertaken on non-dissenting animals. Thus, if an animal expresses dissent, they are not required to participate in research, either at that time or at all. This removes practical challenges generated by dissent as well as epistemological worries involving data being skewed by animal stress. It does, however, mean that certain data might not be collected, as research might not be undertaken if non-dissenting animals cannot be identified. Further, the participation of particular animals may be effectively wasted if participation by conspecifics is insufficient to ensure statistically significant results.

Again, this strategy of only performing protocols on non-dissenting animals differs from the traditional practice of research with animals; it might meet with frustration on the part of researchers keen to undertake particular protocols and uncertainty on the part of AECs over whether to approve protocols in which requisite animal numbers might never be achieved. Although the cessation of research due to an inability to enroll sufficient participants is not new to research with human subjects, it is not a common problem for laboratory-based animal research.

In addition, it is important to consider how animals that have expressed dissent are treated. It may be that their dissent is not sustained, meaning that researchers can attempt to successfully run the protocol later. If the dissent persists, however, such non-compliant animals are unlikely to be permitted to continue living in a research facility. In this circumstance, one option compatible with taking a vulnerability approach seriously would be to rehome the dissenting animal. Shifting to becoming a companion or sanctuary animal likely alters the situational vulnerability of research animals in a positive way. Rehoming sits well with a consequentialist approach too, provided the new home environment meets the animal’s welfare needs.

However, if a dissenting animal is euthanized, different worries arise in a vulnerability approach, as this amounts to creating a form of pathogenic vulnerability. Recall that pathogenic vulnerabilities can be introduced via attempts to address existing vulnerabilities. Responding to dissent by euthanizing the dissenter appears to create just such a situation, whereas on the consequentialist account euthanasia can be deemed a legitimate course of action if triggered by significant compromises to welfare that are not essential to the research protocol. 

Finally, we move to the most significant shift in research provoked by the concern for dissent proposed here. This involves making a prospective assessment about how research may be received by animal participants. If, on such an assessment, it is deemed likely that dissent at some point is unavoidable, then this furnishes a reason not to undertake that research. Toxicology research is an obvious example, where even though participants might enter research willingly, they are likely to rapidly exhibit aversive reactions. Even in a conservative estimate, much research would in fact be captured and ruled out by this proposal.

The practical challenges that attach to dissent are obviously removed under this strategy, though arguably the epistemological challenges merely take on a different form. With research not proceeding, no knowledge is gained. However, bearing in mind that this only applies to research for which there is unavoidable dissent, and as such in many cases the potential for results to be unreliable, this appears less problematic.

It might be objected, however, that the strategies discussed in this section fall prey to the same criticisms made against training, namely, that new pathogenic vulnerabilities are generated. After all, the practice of animal research continues, just as is the case with training. The objection might run that although such changes to research might appear to address ethical issues with animal research, essentially they simply facilitate the status quo. However, this seriously underestimates the difference between circumventing dissent through training and being responsive to the dissent manifested and expressed by animals in research. The latter set of strategies foster a transformed practice of research, one in which researchers attend to the behaviour of individual animals in a more focused way and tailor their research accordingly.

## 5. Conclusions

Consent, assent, acquiescence, and dissent are underexplored in the context of biomedical research involving animals. This paper has focused on dissent, in part because although certain nonhuman animals may have the capacity to provide their assent to research, strategies that are responsive to dissent can be applied more broadly. Appealing to dissent can work not just for animals such as chimpanzees, but for rodents too, whose numbers make up the bulk of animals in research. How better to describe the fact that mice learn to back themselves into corners so that they cannot be extracted from their cage for research [35], than in terms of dissent, a refusal to participate in protocols? Determining the details of how dissent manifests in different animals and how it can be implemented requires further empirical investigation. That said, being attentive to animals, attempting to discern the reactions of individuals to research participation, and responding appropriately to this has a number of payoffs, including the potential to deliver a more epistemically and ethically robust practice of animal research.

## Data Availability

Not applicable.

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
