# Peer review of "A New Strategy for Animal Research: Attending to Dissent"

_animals, 2023, doi:10.3390/ani13091491_

Round 1

Reviewer 1 Report

The paper explores the concept of dissent in animal research, it's practical, epistemological, and ethical relevance, and four options with regard to dissent for researchers using animals: ignore, override, training to avoid, and altering research. (The authors organization only identifies the last 3 as options and treats ignoring differently, although it's not clear why.) The topic is important and timely and falls within the scope of the journal Animals.

The novel part of the article is the analytic distinction between the four different responses to dissent and the discussion of how well each response addresses the reasons for not ignoring dissent. However, the conclusions drawn here are vague and general, listing pros and cons with each response, but that's about it. It is not clear to me that they are of enough interest to merit publication. 

Reviewer 2 Report

This is a very interesting addition to the ‘use of animals in research’ ethical debate.

I just have a couple of suggestions that may add new angles to the paper. The first is deeper consideration of advocacy. You touch on this in Section 4 (lines 204-6) and in Section 4.3, where you mention that research workers need to pick up on signs of dissent to enable the process of removing dissenting animals from research protocols. However, it might be useful to explore the differences between the laboratory situation and veterinary clinical research on owned animals, where each animal subject has an attached human participant who can consent on their behalf, and would be regarded as their advocate. Would ‘Named Animal Care and Welfare Officers’ (required in the UK under research legislation), or equivalent in the various jurisdictions, have similar responsibilities for advocacy in research institutes and could they be seen as suitable people to make decisions regarding dissent?

The second suggestion regards the fate of dissenting animals. I think there is an ethical discussion to be had over whether being killed is preferable to living in a research laboratory, because euthanasia/killing would be the fate of the dissenting animal. Without considering this, the bigger picture debate over whether animals should be used in research pales into insignificance. These animals are in that situation now, and we must care about what happens to them.

I hope that these are constructive suggestions, and would be keen to hear your response.

Reviewer 3 Report

Review A New Strategy for Animal Research: Attending to Dissent

The concept of dissent in this context is not new, see Kantin and Wendler 2015 as cited but this is an interesting wider development of this approach and providers useful challenge.

The introduction commences with reports on the success rate of translation of animal studies to successful intervention sin people. Leaving aside there are other, non animal related reasons for such failures, this seems an unnecessarily narrow premise of this paper’s ‘impetus to change’.

There is first the question of what is the alternative, i.e. if this risk of failure is not accepted, what treatments might be denied to people? Is there then public acceptance of reducing access to medicines.

Then there is the more specific issue of Alternatives (as in the 3Rs), why they cannot be used ( for valid scientific  or regulatory reasons) or why they are not being used (inertia, good-think, lack of knowledge etc.).

A. The author might like to consider a wider perspective in their introduction, with thoughts on recent changes in the use in the requirements for regulatory acceptance (especially in the USA), and the recent thoughtful analysis from Marshall et al https://doi.org/10.3390/ani12070863 on barriers to Alternatives (as well as the authors own cited work).

B. I was also surprised that consideration was not given to how dissent is utilised as an approach in studies involving children and those with dementia or other similar cognitive impairments, and what insights could be gained.

Some previous work in the area pertains to chimpanzees, and it is precisely because of their evolutionary closeness that there perhaps ought to be some discussions on the cautions in transferring this approach to all animals, e.g. Turner 2019 https://doi.org/10.1080/02580136.2019.1691359 

C. This issue of applicability from chimpanzees, to rodents, to perhaps even the now very widely used zebra fish should be considered, for both similarities and differences.

The big challenge in utilising and being responsive to dissent is well summarised by the author:

“To be responsive to dissent, its presence needs to be detected, and as indicated above determining whether it is genuinely present is challenging and the process fallible. What identifying dissent demands of research animal workers is a set of skills which may be outside their ken, and a potentially harmful attentiveness to research animals “

The author’s example involving dogs illustrate this challenge well:

‘The research involved taking functional MRIs of the dogs, and during operation of the scanner, dogs were unrestrained and thus able to exit at will. Assent was regarded as demonstrated when an animal did not exhibit ‘any signs of not wanting to participate’ .

One could then speculate that a dog might leave the MRI for a range of reasons from an (inaudible to humans) aversive ultrasonic noise, to hearing its caregiver and wanting to join them. Some reasons of apparent non-participation may not be dissent?

D. So I think some consideration is need to given to how dissent might be utilised in practice. This would not be a treatise on implementation, but to recognise both how recent progress can be used but also illustrate we must better understand animals to effectively utilise dissent. As an example, only in the last decade have grimace responses become understood: Mota-Rojas et al 2020 https://doi.org/10.3390/ani10101838 

I do then note this in the discussion:

‘Even though training may be a well-intentioned mechanism for achieving the good end of limiting fear, anxiety and stress in particular research animals, nonetheless it may worsen the situation of research animals in general, by facilitating the status quo. With appropriate training in place, the practice of research can continue; suffering is minimised and dissent is not expressed. Thereby training, through shifting attention away from what might otherwise be in the foreground, may forestall a deeper potentially transformative questioning of research’.

This submission appears, from its content and submission choice to be directed at researchers, who in turn may hold somewhat contrary views, and to make the point, be perceived by them as damned if you do and damned if you don’t.

E. I am afraid this could come across as somewhat of a perceived apparent underlying bias, paradoxically forestalling the desired deeper potentially transformative questioning of research. This which rather undermines what is otherwise an important challenge. I do recognise this is somewhat addressed in lines 340-349, I would simply suggest some reflection on this point to facilitate engagement with researchers on this topic.

Round 2

Reviewer 3 Report

thank you for the changes